# Acute Flaxseed Intake Reduces Postprandial Glycemia in Subjects with Type 2 Diabetes: A Randomized Crossover Clinical Trial

**DOI:** 10.3390/nu14183736

**Published:** 2022-09-10

**Authors:** Fernanda Duarte Moreira, Caio Eduardo Gonçalves Reis, Alexis Fonseca Welker, Andrea Donatti Gallassi

**Affiliations:** 1Coordenação Geral de Atenção às Pessoas com Doenças Crônicas, Ministério da Saúde, Brasília 70058-900, Brazil; 2Programa de Pós-Graduação em Ciências e Tecnologias em Saúde, Universidade de Brasília, Brasília 72220-275, Brazil; 3Departamento de Nutrição, Universidade de Brasília, Brasília 70910-900, Brazil

**Keywords:** dietary fiber, diabetes mellitus, flaxseed, postprandial blood glucose, complex carbohydrates

## Abstract

Background: Postprandial glycemic excursions are associated with impairment control of diabetes mellitus. Long-term consumption of flaxseed can lower blood glucose levels; however, its effects on the postprandial glycemic response remain unknown. Therefore, this study aimed to evaluate the acute effects of raw flaxseed consumption on the 2 h postprandial glycemic curve in men with type 2 diabetes mellitus (T2DM). Methods: This was a randomized crossover clinical trial. Nineteen men with T2DM were randomly assigned a standardized breakfast without (control) or with a previous intake of 15 g of ground raw golden flaxseed (flax). Glycemia was measured at fasting and postprandial at 15, 30, 45, 60, 90, and 120 min. Palatability markers (visual appeal, smell, and pleasantness of taste) and taste intensity (sweetness, saltiness, bitterness, sourness, and creaminess) were evaluated. Results: The peak glucose rise and the 2 h AUC glycemic response reduced in the flax group by 17% (*p* = 0.001) and 24% (*p* < 0.001), respectively. The glucose peak time, palatability, and taste parameters did not differ between the two groups. Conclusions: Ingestion of 15 g of ground raw golden flaxseed before breakfast decreases the 2 h postprandial glycemic response in men with T2DM.

## 1. Introduction

There is extensive evidence that the postprandial glucose peak is associated with diabetes mellitus and its complications [1]. The greater the postprandial glycemic excursion, the greater the production of reactive oxygen species (ROS), oxidative stress [2], protein damage [3], the glycosylated hemoglobin level [4], the glycated albumin concentration [5], and hepatic de novo lipogenesis [6]. Treatments that promote the attenuation of postprandial hyperglycemia observed in individuals with type 2 diabetes mellitus (T2DM) improve several parameters, such as inflammation, endothelial dysfunction, cardiovascular damage, atherogenesis predictors, and aging indicators [7].

Some foods, such as flaxseed, have antihyperglycemic properties without severe reported side effects [8,9]. Flaxseed (*Linum usitatissimum* L.) contains dietary fiber (33.5%; soluble:insoluble fiber = 30:70), lipids (32.3%), proteins (14.1%), carbohydrates (9.8%), and phenolic compounds [10,11]. The functional compounds of flaxseed, such as alpha-linolenic acid (omega 3), fiber (soluble and insoluble), and phenolic compounds (phenolic acids, lignans, flavonoids, and tocopherols) provide several health benefits related to disease improvement in individuals with metabolic syndrome [8,12,13,14]. Long-term consumption of flaxseed (for 8 to 12 weeks) reduced blood glucose, glycated hemoglobin [15,16], triglycerides, total cholesterol, and blood pressure in patients with T2DM [16] and decreased blood glucose and insulinemia and improved insulin sensitivity in individuals with prediabetes and with overweight or obesity [13]. Acute consumption of approximately 30 g of flaxseed, added to a bread recipe or in a 50 g glucose challenge, reduced the blood glucose area under the curve for over 120 min in healthy young adults [17,18].

Several studies have shown that flaxseed consumption improves glycemic control [13,15,16,19]. However, to date, there are no studies that have investigated the acute effects of flaxseed on postprandial hyperglycemia in individuals with T2DM. Therefore, this study aimed to evaluate the acute effects of the consumption of ground raw golden flaxseed on the excursion of postprandial hyperglycemia in individuals with T2DM, induced by a meal containing complex carbohydrates. The study hypothesis was that the consumption of 15 g of flaxseed immediately before breakfast could reduce the 2 h postprandial glycemic response in subjects with T2DM.

## 2. Materials and Methods

### 2.1. Participants

Study participants were recruited through public announcements in the public health system (Brasília/Brazil). Eligibility criteria included the following: male subjects with T2DM, age 30–60 years, regular breakfast consumers (≥100 kcal ingested within 2 h of waking for ≥4 d/week), willingness to eat all test foods, no self-reported allergy to the foods provided in the study, and no self-reported sleep disorders. Patients’ medical charts were checked to identify and exclude those with any other clinically diagnosed health conditions. In addition, participants on current exogenous use of insulin, with health complications from diabetes, reported gastrointestinal disorders, and irregular intestinal rhythm (diarrhea or constipation), and smokers were excluded.

Sample size calculation was performed using G*Power software (version 3.1.9.2; Dusseldorf University, Dusseldorf, Germany) [20], considering a superiority margin of 1, a true mean difference in glucose of 5 mmol/L min (or 90 mg/dL min), and a standard deviation of 5. For achieving 80% power at the 5% level of significance, the sample size for a crossover design was 19 participants [21].

### 2.2. Study Design

From June to November 2019, participants were recruited, screened, and randomized into this crossover clinical trial, which required the participants to complete two experimental sessions. Each participant randomly ate the standardized breakfast without (control) and with (flax) the addition of flaxseed.

The clinical trial was registered (https://ensaiosclinicos.gov.br/rg/RBR-98tx28b; accessed: 12 November 2021), and the study protocol was approved by the Human Research Ethics Committee of the University of Brasília (no. 3.317.490) and by the Ethics Committee of the Health Science Teaching and Research Foundation–FEPECS/SES/DF (no. 3.367.200) (Brazil) according to the Declaration of Helsinki statement. All participants were informed about the study and signed written informed consent.

### 2.3. Screening

At the screening visit, participants were asked to complete a screening questionnaire about their diabetes diagnosis, current medications, health conditions, and eating and sleeping habits. In addition, anthropometric measurements and body composition were assessed. Next, participants were instructed to attend the clinical laboratory (06.30–09.30 a.m.), with 8–12 h of overnight fasting without taking any diabetes medication in the morning, for venous blood collection (3 mL). Glucose and insulin were measured by using the glucose oxidase method (ADVIA, 208 model 2400, Siemens Healthcare Diagnostics S.A., São Paulo, Brazil) and electrochemiluminescence (ADVIA, model Centaur, Siemens Healthcare Diagnostics S.A., São Paulo, Brazil), respectively. The sensitivity of glucose oxidase was <5 mg/dL (within-run CV of 3.25%) and of insulin immunoassay was <3.472 pmol/L (within-run CV of 6.0%). Furthermore, glycated hemoglobin (HbA1c) was determined by turbidimetric inhibition immunoassay (COBAS, model C 513, Roche Diagnostica LTDA., São Paulo, Brazil), with a sensitivity of <4.2 % and a within-run CV of 1.6%, and the homeostatic model assessment of insulin resistance (HOMA-IR) was performed to evaluate insulin resistance [22].

### 2.4. Anthropometric and Body Composition Measurements

Bodyweight was assessed using an electronic platform scale (Ramuza, Santana de Parnaíba, Brazil) with a capacity of 200 kg and a precision of 50 g. Height was measured using a stadiometer (Balmak, Santa Bárbara d’Oeste, Brazil) fixed to the wall, measuring to the nearest 0.1 cm. The body mass index (BMI) was computed and classified according to the World Health Organization [23]. Waist circumference was measured midway between the lowest rib and the iliac crest, with a precision of 0.1 cm, and the body fat percentage was measured using multi-frequency bioelectrical impedance InBody 570 (InBody, Seoul, South Korea) according to the manufacturer’s instruction.

### 2.5. Experimental Protocol

After abstaining from exercise and alcohol for 24 h, fasting for 8–10 h, and being asked not to take any diabetes medications, 22 men with T2DM attended the clinic in the morning (between 7–10 a.m.) twice, with a 3–10-day washout period between visits. Each participant randomly (simple randomization using an Excel spreadsheet) ate the breakfast within 15 min either without (control) or with (flax) prior consumption of 15 g of ground raw golden flaxseed (*Linum usitatissimum* L.). The participants consumed 150 mL of water with or without flaxseed immediately before breakfast. They were allowed to take their diabetes medications at the end of each experiment (after completing all blood draws).

The standardized breakfast meal (control) was balanced according to recommendations [24], providing 337 kcal, composed of 50 g of complex carbohydrates (59.4%), 10 g of proteins (11.9%), 10.7 g of lipids (28.7%), and 2.4 g of fiber (200 mL of peach juice, 40 g of toast, and 40 g of cheese).

In the flaxseed intervention (flax), the same standardized control breakfast was eaten immediately after the 15 g portion of the ground raw golden flaxseed, which contained 4.8 g of lipids (3.6 g of polyunsaturated fat, 0.8 g of monounsaturated fat, and 0.4 g of saturated fat), 1.5 g of carbohydrates, 2.1 g of protein, and 5 g of dietary fiber (3.5 g of insoluble fiber and 1.5 g of soluble fiber) [10].

### 2.6. Blood Glucose Assessment

The glycemic response was measured through capillary finger-stick blood samples using the Accu-Chek Active glucometer (Roche, São Paulo, Brazil), according to ISO 15197 requirements (r = 0.998 with the hexokinase method) [25], at fasting and at 15, 30, 45, 60, 90, and 120 min postprandially. The 2 h incremental area under the glycemic response curve (AUC) was calculated using the trapezoidal method, excluding the values below the baseline [26].

### 2.7. Palatability Assessment

Participants were asked to complete a visual analog scale (VAS) to assess palatability parameters (visual appeal, smell, and pleasantness of taste) and taste intensity (sweetness, saltiness, bitterness, sourness, and creaminess) 15 min after breakfast [27]. An 11-point scale, with 0 being bad and 10 being good, was adapted to assess breakfast, and ratings were recorded, with higher numbers indicating greater pleasantness.

### 2.8. Statistical Analysis

Data were tested for normality using the Shapiro–Wilk test and for homoscedasticity using Levene’s test. The glucose peak rise and 2 h glycemic AUC data were compared between flax and control groups using paired *t*-tests. A non-parametric test compared the time-to-glucose peak using the Wilcoxon test. Palatability score data were compared between flax and control groups using the paired *t*-test or the Wilcoxon test, according to the test result for normality. The postprandial glycemic response curve was analyzed using two-way ANOVA repeated measures with the Bonferroni test for post hoc comparisons. The values were expressed using means ± standard deviations (SDs). The analyses were conducted using the Statistical Package for the Social Sciences (SPSS version 21.0) by adopting statistical significance criteria of *p* < 0.05.

## 3. Results

The screening visit was attended by 93 subjects, and 19 participants completed the full study protocol (Figure 1). All participants were currently using oral hypoglycemic medications—9 (47.4%) used only one drug (metformin = 8 and gliclazide = 1); 5 (26.3%) combined metformin and glibenclamide; 2 (10.5%) associated metformin with vildagliptin; 2 (10.5%) combined metformin, glibenclamide, and dapagliflozin; and 1 (5.3%) used metformin, glibenclamide, and pioglitazone. There was no change in the type and/or dosage of medications during participation in the study. The participants’ characteristics are shown in Table 1.

The participants’ fasting blood glucose did not differ between flax and control groups (*p* = 0.35). Fasting blood glucose was measured immediately before eating the tested meals as a baseline control to confirm that there were no differences between flax and control groups that could influence the assessed glycemic response. The glucose peak rise was 87 mg/dL after the intake of the control breakfast (from 128.0 ± 5.5 mg/dL at fasting to 215.3 ± 7.2 mg/dL at peak). Consumption of 15 g of flaxseed reduced the glucose peak rise by 17.25% (133.0 ± 6.2 mg/dL at fasting to 205.1 ± 8.4 mg/dL at peak; *p* = 0.001; Figure 2A). The intake of flaxseed before breakfast also decreased (23.99%) the 2 h glycemic AUC (*p* < 0.001; Figure 2B) and the postprandial glycemic response at 30, 45, and 60 min compared to the control groups (time-meal interaction *p* < 0.001; Figure 2C). There was no difference in the time-to-glucose peak (*p* = 0.30) between flax and control groups.

There were no significant differences in the palatability scores for visual appeal (*p* = 0.89), smell (*p* = 0.96), and pleasantness of taste (*p* = 0.35) between flax and control groups. In addition, no differences were found in the taste intensity of sweetness (*p* = 0.48), saltiness (*p* = 0.91), bitterness (*p* = 0.27), sourness (*p* = 0.42), and creaminess (*p* = 0.72) between the tested meals. Therefore, the addition of the flax did not affect the overall palatability of the standardized breakfast (Table 2).

## 4. Discussion

Ingestion of 15 g of ground raw golden flaxseed immediately before a breakfast containing complex carbohydrates decreased the glycemic 2 h AUC response by 24%, the glucose peak rise by 17%, and the postprandial glycemic response at 30, 45, and 60 min compared to the control group in men with T2DM. In addition, the palatability of the meal was not altered by previous ingestion of raw flaxseed, corroborating the consumption of flaxseed as a viable strategy that can be applied in the daily life of individuals with T2DM to reduce postprandial hyperglycemia.

Several studies have shown that the glycemic response in individuals with prediabetes or T2DM is reduced by long-term consumption (4 to 12 weeks) of flaxseed [13,28,29,30]. However, to date, few studies have investigated the acute effects of eating flaxseed before or during a meal. One study evaluated the acute postprandial glycemic response in 15 adults with T2DM after they ate puddings containing yellow mustard mucilage, fenugreek gum, and flaxseed mucilage. The blood glucose and plasma insulin at peak concentration and specific time points significantly decreased after consumption of all soluble fiber puddings compared to the control pudding but did not differ from each other [31]. Another study compared the effects of two seeds on postprandial glycemia and satiety scores. Fifteen healthy participants were randomized to receive a 50 g glucose challenge, alone or supplemented with either 25 g of ground Salba chia or 31.5 g of flaxseed, on three separate occasions. Blood glucose samples were collected at fasting and over 2 h postprandially. Both Salba chia and flaxseed reduced the glycemic 2 h AUC response compared to the glucose control [18]. Moreover, most clinical trials that showed that flaxseed reduces the acute glycemic response in healthy people used approximately a 30 g serving [17,18,32]. Our work has shown that just 15 g of ground raw golden flaxseed, consumed immediately before a meal (no need to be added to culinary preparations), is sufficient to reduce the postprandial glycemic response. Furthermore, our results were observed in subjects with T2DM, a population that could particularly benefit from these findings.

The glycemic reduction observed in studies that have evaluated the long-term effects of flaxseed consumption is often attributed to its soluble fiber content. The role of soluble fiber in the digestion process is well described in the literature, and the main related mechanisms include slower gastric emptying [33], greater volume and viscosity of the food bolus [34], and delay in the interaction between digestive enzymes and nutrients [35], which delays the breakdown of complex nutrients into absorbable components and slows the absorption of glucose at the brush border, resulting in a curve with a lower glycemic peak [34,35]. The mechanisms by which insoluble fiber may improve glycemia have been discussed in the literature, although they are still under-investigated and not fully understood. Some possible explanations include interference of insoluble fibers with the absorption or digestion of dietary protein and the promotion of fecal excretion of branched-chain amino acids, reduction in the hepatic glucose output, an increased rate of transit through the small bowel, inhibition of enzymes related to the digestion of complex carbohydrates, and alteration of the composition of gut microbiota [36,37,38].

The portion (15 g) of raw flaxseed tested in this study provides a low amount of soluble fiber (1.5 g) in relation to the insoluble fiber content (3.5 g). This reinforces the hypothesis that the type of fiber may be less relevant than the total amount of fiber to promote some health benefits [39] or even that the dose of insoluble fiber contained in 15 g of flaxseed is sufficient to inhibit the enzymes (alpha-glucosidase/alpha-amylase) responsible for carbohydrate digestion [40].

It is worth noting that flaxseed protein may also help in the prevention and treatment of heart disease and support the immune system. In addition, flaxseed is the richest plant source of the ω-3 fatty acid, i.e., α-linolenic acid, which is also essential in the prevention and treatment of diabetes, atherosclerosis, cardiovascular diseases, dyslipidemia, and metabolic syndrome [41]. However, in the dose of 15 g of flaxseed, the amount of protein (2.1 g) and fat (4.8 g) may be insufficient to influence the postprandial glycemic response [10,11]. According to the literature, >15 g of fat is generally required for a significant decrease in the glycemic response to foods, when at least 50 g of carbohydrates is consumed [42,43,44].

The physiological fiber response can be altered by the subject’s intestinal rhythm [45]. This variable was not controlled in a study that evaluated the effects of flaxseed on the glycemic control in well-controlled individuals with T2DM [46]. In our study, we asked the participants about their intestinal rhythm, and those who reported irregular bowel frequency, such as diarrhea or constipation, were not enrolled in the study protocol.

Furthermore, we performed the interventions under standardized conditions and in a real-life context, where patients had moderate chronic glycemic control (mean HbA¹c ± 7.0%), fasting glucose levels were similar in all experimental sessions, and meals were consumed during a similar period (up to 15 min). In addition, in contrast to another study that assessed the glycemic response after glucose ingestion (50 g glucose challenge) [18], our study evaluated the postprandial glycemic response after the addition of flaxseed to a nutritionally balanced meal in macronutrients with complex carbohydrates, which is better suited to the real-life context of patients with T2DM [47].

Despite the relevant results of raw flaxseed in the postprandial glycemic response, it is important to consider that regardless of health-related claims and recommendations, taste, texture, appearance, color, and aroma will play an important role in people’s choice for their usual food consumption [8]. In our study, the taste intensity (sweetness, saltiness, bitterness, sourness, and creaminess) and palatability scores (visual appeal, smell, and pleasantness of taste) of both meals were considered good by the participants, with no significant difference between flax and control groups. Thus, the consumption of 15 g of raw flaxseed with 150 mL of water immediately before a meal did not change its sensory characteristics and palatability, making this dietary approach simple, accessible, and feasible for patients with T2DM who need to improve their glycemic control.

Therefore, the evaluation of the postprandial glycemic response in a real-life context provides clinical applicability to our results, as the effects of raw flaxseed consumption were observed in individuals with T2DM, which corresponds to the largest part of people with diabetes in the world. Thus, as these patients may benefit from a nutritional strategy aimed at improving postprandial glycemic excursions (typically high in these individuals), the results presented might contribute to decreasing the 2 h postprandial glycemic curve, leading to better glycemic control [47].

This is the first study to investigate the acute effects of ground raw golden flaxseed consumption on the 2 h postprandial glycemic response in men with T2DM. Before this study, the effects of a single dose of raw flaxseed on postprandial glycemic excursions in men with T2DM were still unclear. However, our study had some limitations. The exclusion of patients with diabetes using exogenous insulin or with micro- or macrovascular complications makes the results of this study limited to the characteristics of this sample and should be extrapolated with caution. The intervention was not blinded, and therefore, there is a potential risk of expectations influencing the findings, most obviously when there is some subjectivity in the assessment, such as the palatability questionnaire. In addition, our study did not include women due to the need to monitor hormone levels, particularly at menopause, as it is related to changes in insulin sensitivity and carbohydrate metabolism [48], which could be a potential confounding factor. Therefore, the presented results should not be generalized to women. In addition, the presented results should be applied with caution due to the acute effect experimental design, since long-term studies are needed to confirm the effects of flaxseed consumption on the postprandial glycemic response in the T2DM population to support a strong clinical recommendation for use.

## 5. Conclusions

The study results indicate that ingestion of 15 g of ground raw golden flaxseed before a breakfast containing complex carbohydrates decreases the 2 h postprandial glycemic response in men with T2DM compared to a control breakfast (no flaxseed ingested). Our findings have practical implications as golden flaxseed is wildly available in many countries and represents a simple and affordable dietary strategy to be adopted by individuals with T2DM to improve glycemic control. We recommend that for future studies, glucose and insulin levels, beta-cell function, insulin sensitivity, and incretin hormones should be assessed to understand the mechanism of action of flaxseed in reducing postprandial glycemic excursions.

## Figures and Tables

**Figure 1 nutrients-14-03736-f001:**
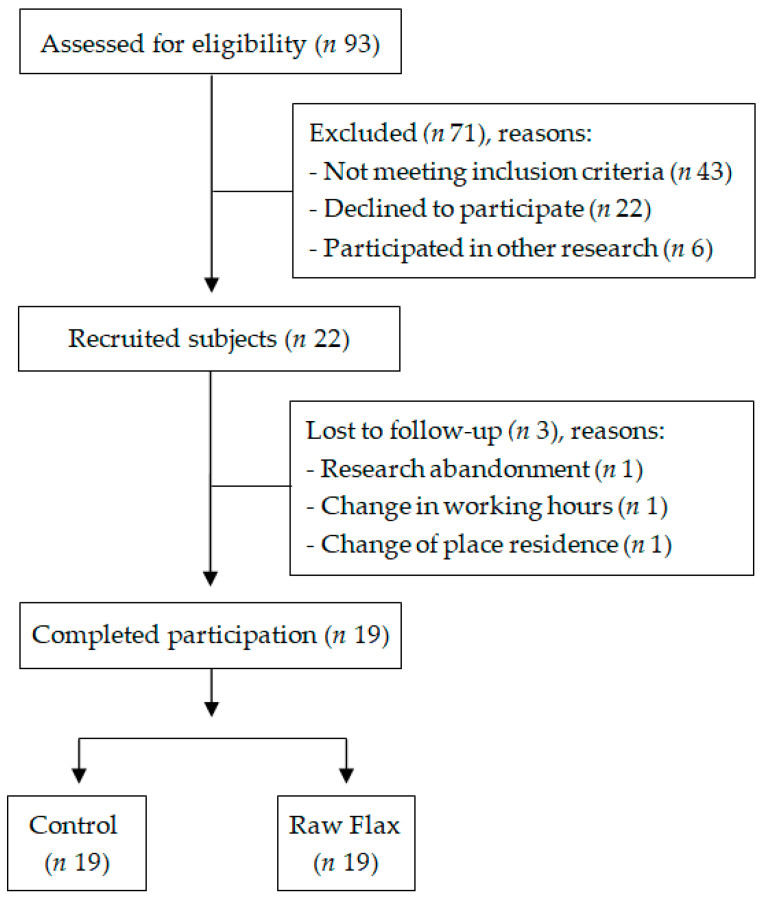
Study participants’ flowchart (*n* = 19).

**Figure 2 nutrients-14-03736-f002:**
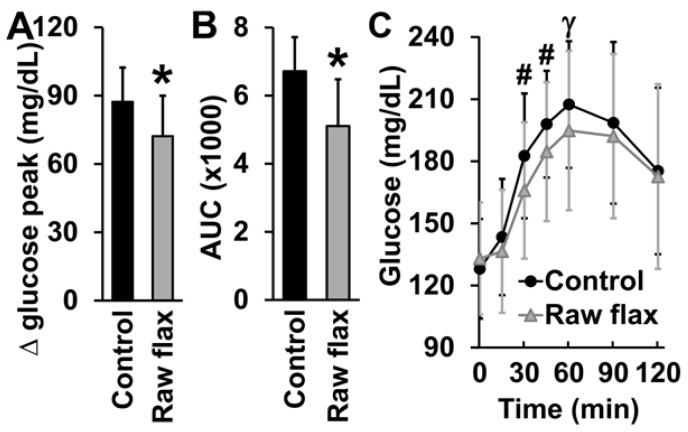
Postprandial glycemic responses of men with T2DM in the control and raw flaxseed groups. (**A**) Glucose peak rise in men with T2DM after ingesting a breakfast containing 50 g of complex carbohydrates either without (control) or with (flax) prior intake of 15 g of raw flaxseed, (**B**) area under the glycemic curve (AUC; mg/dL/min), and (**C**) glycemic curve. * *p* ≤ 0.001 vs. control; # *p* = 0.03 vs. control; γ *p* = 0.072 vs. control.

**Table 1 nutrients-14-03736-t001:** Baseline characteristics of study participants (*n* = 19).

Variables	Mean ± SD	Range
Age	52.1 ± 6.7	38–59
Body weight (kg)	94.0 ± 17.5	67.6–128.4
Height (cm)	172.5 ± 7.9	154–192
BMI (kg/m²)	31.7 ± 5.5	23–41
Waist circumference (cm)	109.1 ± 12.7	85–132
Body fat (%)	31.3 ± 9.1	15.6–46.6
Diabetes duration (months)	76.5 ± 72.6	6–300
Fasting blood glucose (mg/dL)	124.7 ± 26.0	81–191
Insulin (µUI/mL)	15.6 ± 8.1	4.0–33.7
HbA1c (%)	6.9 ± 0.8	5.6–8.3
HOMA-IR	4.8 ± 2.5	1.2–9.7

SD = standard deviation; BMI = body mass index; HbA1c = glycated hemoglobin; HOMA-IR = homeostasis model assessment–insulin resistance.

**Table 2 nutrients-14-03736-t002:** Scores of sensory characteristics of the control breakfast and breakfast with 15 g of ground raw golden flaxseed. Data are expressed as the mean ± SD.

Palatability	Control	Flax	*p*-Value
Visual appeal	7.7 ± 2.0	7.8 ± 1.7	0.89
Smell	7.6 ± 1.7	7.6 ± 1.7	0.96
Pleasantness of taste	7.6 ± 1.7	7.2 ± 1.8	0.35
Sweetness	5.3 ± 2.3	4.6 ± 2.5	0.48
Saltiness	3.1 ± 2.3	3.1 ± 2.2	0.91
Bitterness	2.3 ± 2.2	2.4 ± 2.0	0.27
Sourness	2.4 ± 2.1	2.4 ± 1.9	0.42
Creaminess	7.5 ± 2.1	7.4 ± 2.0	0.72

## Data Availability

Not applicable.

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
