# Peer review of "Acute Flaxseed Intake Reduces Postprandial Glycemia in Subjects with Type 2 Diabetes: A Randomized Crossover Clinical Trial"

_nutrients, 2022, doi:10.3390/nu14183736_

Round 1

Reviewer 1 Report

This study evaluated the acute effect of pre-breakfast consumption of ground flax on post-prandial glucose responses in 19 men with T2DM in a cross-over trial. They found that 15 grams of flax consumed prior to breakfast reduced post-prandial peak and AUC glucose.  

This is a nice pilot study showing acute benefit of flax consumption on post-prandial glycemic. The findings are novel and the study well conducted with a cross-over design.

The difference in glycemia was rather modest at only 17% decrease in peak glucose and AUC by 24% – is this clinically significant? A previous study in well controlled T2DM x 12 weeks found no difference in A1c after chronic treatment (Taylor et al, J Am Col Nutr 2010). Given that this study tested acute effects only and the lack of effect on glycemic control with chronic use in the study by Taylor et al., I do not think this data can support a strong clinical recommendation for chronic use.

While the discussion included data and hypotheses about mechanism for the lower glycemic excursion with flax seed, the study itself did not include measures to investigate any mechanisms, which is a lost opportunity. For future studies I would recommend collecting blood samples at each time point that can be stored and assayed for both glucose and insulin to assess beta-cell function and other measures of insulin sensitivity and possibly also evaluate gastric emptying as a factor. This would have added more depth to the analysis and interpretation of the data regarding what the mechanism of action of flax is on reducing post-prandial glycemic excursions.

Minor comments:

I found this confusing as to whether the flax was consumed 15 minutes before breakfast or the breakfast was consumed within 15 minutes: “Each participant randomly (simple randomization by Excel spreadsheet) ingested a breakfast either without (Control) or with prior consumption of 15 g of ground raw golden flax seed (flax) with 150 mL of water within 15 minutes.” Might rephrase to clarify that the breakfast was consumed within 15 minutes and that the water with or without flax seed was consumed immediately prior to eating breakfast. Also clarify when the diabetes medications were taken - ? after completing all blood draws for the MTT?

The flax seed increased the meal fat content by 50% and tripled the fiber content. Do you think this could have affected gastric emptying and thus the glucose excursion?

Please clearly state the hypothesis in the introduction.

Both men and women should be included in clinical research studies.

Please proof for grammar and spelling.  See line 39-40 Page 1.

Reviewer 2 Report

The adjectives "chronic" and "acute" are cited in the text 4 times each one with different meanings, and "acute" non in opposite of “chronic”. It may be confusing. I think that the terms acute and chronic that have a negative value, are rather to be reserved to differentiate acute or chronic pathologies, than to describe immediate or overall benefits of glycemic balance. It could perhaps be clarified that the effect of flaxseed was present from the first administration and then continued, rather than induced by a continuous intake, as previously described.
